# Health-Related Quality of Life among Elderly Patients in Urban Bangladesh: A Cross-Sectional Study

**DOI:** 10.3390/diseases12090212

**Published:** 2024-09-11

**Authors:** Joynal Abedin Imran, Amal K. Mitra, Marzana Afrooj Ria, Tumpa Mitra, Jannatul Ferdous Konok, Sadia Akter Shuchi, Pradip Kumar Saha

**Affiliations:** 1Department of Physiotherapy, National Institute of Traumatology & Orthopaedic Rehabilitation (NITOR), Dhaka 1207, Bangladesh; 4abedin@gmail.com (J.A.I.); marzana.afrooz@gmail.com (M.A.R.); tannymitra0419@gmail.com (T.M.); jfkonok1998@gmail.com (J.F.K.); sa.sadia.nitor@gmail.com (S.A.S.); dr.pradip1977@gmail.com (P.K.S.); 2Department of Public Health, Julia Jones Matthews School of Population and Public Health, Texas Tech University Health Sciences Center, Abilene, TX 79601, USA; 3Amola Healthcare & Research, Dhaka 1207, Bangladesh

**Keywords:** elderly patient, quality of life, OPQOL-35, BMI, QoL, sleep pattern, Bangladesh

## Abstract

Quality of life (QoL) focuses on a person’s ability to live a fulfilling life. It helps in determining successful aging in the elderly population. Because of a scarcity of information regarding predictors of QoL in the elderly population in developing countries, this study was undertaken to assess the overall QoL and its predictors in the elderly population in an urban setting of Bangladesh. In this cross-sectional study, 275 participants were enrolled by systematic sampling at the outpatient department of a tertiary care hospital in Dhaka, Bangladesh. Data were collected by using a pretested semi-structured standard questionnaire for QoL (OPQOL-35). Pearson’s correlation was used to assess the association between QoL and sociodemographic factors. Multivariate linear regression was conducted to identify predictors of QoL, after controlling for potential confounders. The median age of the participants (*n* = 275) was 65 years (range, 60 to 85; 25th and 75th percentile, 60 and 68, respectively). The majority (78%) of them were married, and 20% were widowed or divorced. The median score of QoL was 113 (25th and 75th percentile, 101 and 124, respectively). Most of the participants had very poor (bad as can be) or poor (bad) QoL. Only 7.3% were observed to have “good” QoL (scores 140 or more). Females had significantly poorer QoL scores compared to males (*p* < 0.001). Age, gender, educational status, smoking, and sleep duration significantly predicted QoL in a multiple regression analysis. In conclusion, the QoL of the elderly individuals in Bangladesh was very poor or poor. Future research should focus on service-oriented interventions, especially targeting women. Counseling elderly people to enhance their quality of life may include improving sleep patterns, healthy diets, regular exercise, and caring for their mental health.

## 1. Introduction

Population aging is now a daunting challenge across the globe. The number of people aged 60 and older is projected to reach nearly 2.1 billion in 2050 from 382 million in 1980 [1]. Compared to the developed world, the demographic transition is more challenging in Asian countries like Bangladesh, China, and India because of the growing population and inadequate social support system for elderly citizens [2]. According to the United Nations projection, Asia will be the largest resident of elderly people, with 38% (one billion) elderly living in China and India alone [2]. In Bangladesh, approximately 8% of the total population is 60 years or older, projected to rise to 11.5% by 2030 [3] and 20.2% by 2051 [4].

Quality of Life (QoL) underscores the importance of community well-being. According to the World Health Organization, QoL refers to how persons view their places in society, taking into account their cultural and societal norms, as well as their personal goals, aspirations, values, and apprehensions [5]. Therefore, QoL is affected by a wide range of factors—from individual health status to socioeconomic factors. Each of these factors does not affect QoL in isolation; rather, they interplay together. However, data on the determinants of the elderly’s QoL are scarce, particularly in developing countries. In a study of elderly people in China and India, the determinants of perceived QoL of elderly people included gender, household income, education, co-morbidity, and social cohesion [2]. In a previous study in Bangladesh, factors associated with low health-related QoL among older citizens included increasing age, being widowed, having a lower level of education, and lower socioeconomic status. In another cross-sectional study [6], health-related factors, such as sleep problems, depression, and health service availability, and social factors, such as religiosity and social support, were correlated with the QoL of elderly people.

Despite a few reports of cross-sectional studies, several gaps are in the literature. For example, studies [2,7,8] attempted to measure elderly people’s perceived QoL but did not quantitatively measure QoL and its impact. Only a few earlier studies [6,7] showed correlations between QoL and individual and social factors. However, no robust analytical methods (such as multivariate analyses) were used to determine the predictive models of QoL scores among this population. To fill out the gap in the literature, we aimed to quantitatively measure the QoL scores of elderly people and determine the predictors of QoL scores in this population.

## 2. Methods and Materials

### 2.1. Study Design and Selection of Samples

This is a cross-sectional study conducted from March 2023 to February 2024 among elderly patients who visited the outpatient department of the National Institute of Traumatology & Orthopaedic Rehabilitation (NITOR), Dhaka, Bangladesh. The eligible patients included people of either gender, aged 60 to 85. The required patients were enrolled (see sample size estimation below) using a systematic random sample procedure, and every alternate patient was selected from among about the 476 patients who visited the outpatient department of the Hospital daily.

### 2.2. Ethical Procedures

The study was conducted following the Declaration of Helsinki, and the protocol was approved by the Ethics Committee of NITOR (Memo No. NITOR-PT-93-2023-03). The study objectives and the procedures were explained before the informed consent was obtained. Participation in this study was completely voluntary, and the participants had the right to withdraw at any time. They were assured about the anonymity, confidentiality of the data, and risks and benefits. No body samples were obtained. The study data were collected by using a standard questionnaire, which is described below.

### 2.3. Questionnaire

Participants were interviewed using a structured and standard questionnaire (OPQOL-35) to collect information on sociodemographic characteristics and physical health status. A copy of the questionnaire is added at the end of Appendix A and Appendix B. Health-related quality of life (HRQoL) was measured using the Likert scale data collected through the questionnaire. The questionnaire had a 5-point score for each of the 35 questions, with a range of “strongly agree” to “strongly disagree” (1–5 scores). The questionnaire was originally developed by Ann Bowling in 2009 [9]. The total score ranges from a minimum of 35 to a maximum of 175. The total scores were categorized as follows: less than 99 of QoL as bad as can be, 100–119 as bad, 120–139 as optimum, 140–159 as good, and 160–175 of QoL as good as can be [9]. In our study, the survey instrument was translated into Bengali by forward and backward translation, and the validity of the questionnaire was field-tested on 30 elderly participants from the same cohort. Any inconsistencies in data were identified and corrected at the field level. Cronbach’s alpha for reliability was calculated for the 35 items of the QoL questionnaire. The overall Cronbach’s alpha score was 0.88, meaning the QoL questionnaire was reliable.

In addition to QoL-related data, we collected demographic data (including age, sex, marital status, occupation, monthly income, and any special allowances), food habits (especially the daily intake of fruits and vegetables), amount of water intake, personal habits (such as smoking and exercise), and sleep pattern (hours of sleep and any sleep disturbances), and we measured the weight and height of individuals and calculated body mass index (BMI).

### 2.4. Sample Size Estimation

The major variable of interest in QoL is the overall life satisfaction of the individuals. We used data on patient satisfaction from Khan and colleagues [10] for the sample size estimation. Using the data, we calculated the sample size as follows:$$n=\frac{Z^{2}pq}{d^{2}}$$

Here, the proportion of the people who had satisfaction, *p* = 20.4% or 0.204

*q* = 1 − *p* = 0.796

*Z* = 1.96 for α = 0.05 (95% confidence)

The required sample size was 275 after adjusting for an additional 10% attrition rate.

### 2.5. Statistical Analysis

The data analysis was conducted using IBM SPSS version 29. First, a descriptive statistical analysis was performed to assess data distribution. Bivariate correlations were assessed between QoL and sociodemographic factors. For the multivariate analysis, a linear regression model was constructed with the QoL score as a dependent variable. Only the covariates that were significantly associated with QoL in the bivariate analyses were included in the multivariate model. Statistical significance was determined using a *p*-value ≤ 0.05.

## 3. Results

### 3.1. Sociodemographic Characteristics

In this study, 275 adults were enrolled, and none dropped out. The age of the participants ranged from 60 to 85 years, with the mean ± *SD* of age being 65 ± 5.5 years (95% confidence intervals = 0.69 to 3.17). Only 9% (*n* = 25) were aged 75 years or older. The majority of the participants were male (62.9%), married (78.2%), and had no formal education (41.8%) or primary education (19.3%) (Table 1). Based on the monthly family income, 75.3% were considered low-income families. About 75% had no additional income in terms of government-supported allowances, such as freedom fighter allowance, old-age allowance, or disability allowance.

The mean age of males was somewhat higher than that of females (*p* = 0.002). There were also significant gender differences in marital status (*p* < 0.001), educational categories (*p* < 0.001), and occupation (*p* < 0.001) (Table 1).

### 3.2. Difference in Characteristics of Physical Health and Diet

Table 2 presents characteristics related to physical health, diet, smoking, sleep pattern, and QoL of the population. The BMI was normal in the study participants, with a mean ± *SD* of 24.36 ± 3.32. The obesity rate was higher in females (10.8%) than in males (4.0%). More than 75% of the people ate vegetables either daily or often. Similarly, about 70% ate some form of protein-rich foods, such as fish, meat, eggs, or lentils, daily or often. However, fruit consumption was less common, comprising only 12% daily and about 15% often having fruits. As smoking is not socially acceptable in females in Bangladesh, none of the female participants reported ever smoking, whereas 50% of the men were current or former smokers. The majority of individuals (51.6%) reported having 6 to 7 h of sleep per day, which was quite adequate.

### 3.3. Quality of Life Scores

The study subjects (*n* = 275) had an overall poor QoL score, as indicated by the mean ± *SD* scores of 113.79 ± 16.50. Among them, females had significantly poorer scores of QoL compared to their male counterparts (109.33 ± 16.06 vs. 116.41 ± 16.23, *p* < 0.001). When the categories of QoL were compared, most of the participants (65%) had very poor to poor QoL scores (bad as can be or bad), whereas over a quarter of them had the optimum level of scores, and only about 7% had good scores of QoL.

### 3.4. Distribution of Elderly People by Quality of Life (QoL) Categories

Among the 275 participants, the highest score observed was 159, while the lowest score was 70 out of a total possible score of 175. The median score of QoL was 113, while the 25th and the 75th percentile of the scores were 101 and 124, respectively. A significant proportion of the participants (65%) were identified as having a very poor (bad as can be) or poor (or bad) QoL. Only 7.3% were observed having “good” QoL. None of the participants had the highest score of having “QoL good as can be”.

### 3.5. Correlation between Sociodemographic Factors and Quality of Life (QoL) Scores

A bivariate correlation test (Table 3) showed that QoL directly correlated with educational status (*r* = 0.417, *p* <0.001), BMI (*r* = 0.159, *p* = 0.008) and hours of sleep (*r* = 0.195, *p* = 0.001). QoL was inversely correlated with age (*r* = −0.128, *p* = 0.034), meaning poorer QoL with increasing age, even within the elderly population. The correlation coefficient (*r*) value of the association between QoL and gender was negative, meaning that the QoL scores were higher in males than in females (gender: male = 0, and female = 1).

### 3.6. Linear Regression Analysis to Predict Quality of Life

The statistically significant variables in bivariate analysis were entered into the regression analysis model. The dependent variable was the QoL score, which was a continuous variable. The selected independent variables were age, gender (0 = male, 1 = female), marital status (0 = married, 1 = other), educational status (0 = no formal education, 1 = any education), smoking status (0 = never smoker, 1 = smoker), and sleep duration (hours). Table 4 shows the results of a stepwise multiple linear regression analysis. The statistically significant predictors of QoL scores included age (*p* = 0.016), gender (*p* < 0.001), educational status (*p* < 0.001), smoking status (*p* = 0.001), and sleep duration (*p* = 0.003). Marital status was not a significant predictor (*p* = 0.192).

The predicted model for the data was as follows:

Quality of life (scores) = 128.39 − 0.408 age − 9.202 gender (0 = male, 1 = female) + 8.298 educational status (0 = no formal education, 1 = any education)) − 7.426 smoking status (0 = never smoker, 1 = smoker) + 2.101 sleep duration.

## 4. Discussions

In this study, elderly people showed poor QoL scores. Females had worse scores than men. QoL was significantly correlated with the sociodemographic parameters studied. Multivariate analysis also indicated that educational status, smoking, sleep duration, and marital status significantly predicted quality of life scores.

The demographic shift of the population due to increasing longevity, together with the epidemiologic transition from infectious and contagious diseases to chronic and non-communicable diseases, raises great concern for the quality of life of the increasing number of elderly people in any society. Consequently, there is a growing trend in appraising the quality of life among elderly populations using a survey instrument (OPQOL-35) similar to our study [11] and also by using modified survey tools [12]. However, studies showed acceptable levels of reliability and validity in using OPQOL-35 in a multidimensional population, and therefore, OPQOL-35 has a broader use [13]. In our study, the survey instrument was translated into Bengali and field tested, and the instrument’s reliability was strong (with a Cronbach’s alpha of 0.88).

The relationship between age and HRQoL among the elderly varied, with poorer QoL with increasing age. Additionally, in other studies [14], factors such as gender, education, employment status, and smoking also influenced HRQoL in older adults. Increasing age is associated with a decline in health-related quality of life (HRQOL) in the elderly [15,16], as our study showed. Studies have shown that as individuals age, the prevalence of chronic diseases and disorders increases, leading to a decrease in HRQOL [17]. In a study conducted in India, it was found that older age, male gender, lower education, absence of a spouse, lower economic status, and chronic disorders were associated with lower HRQoL [16].

Gender could be a strong factor in determining quality of life. However, studies show conflicting results. In this study, the QoL of female participants was significantly poorer than that of males in both bivariate and multivariate analyses. In Iran, a study on the quality of life of elderly people showed that males lead a better quality of life than females from the perspective of functional capacity, perceived health, good housing conditions, an active lifestyle, economic status, and good social relationships [18]. On the contrary, another study suggested that longstanding illnesses significantly reduce the quality of life in men but not women [19]. The observed poorer QoL in Bangladeshi females is more likely due to societal neglect of them in general in the country, which leads to females living in more vulnerable health conditions and healthcare that adversely affect their QoL.

Marital status has been found to have a significant impact on the QoL of elderly individuals. Being married is associated with higher levels of life satisfaction and better QoL in terms of physical and psychological health and social relationships [20,21]. Similarly, in this study, being married was a positive predictor of the overall QoL of elderly people. As socioeconomic problems are more common in older age, they have no scope to share their emotion and feelings when a partner is absent.

The educational qualifications of elderly individuals play a crucial role in shaping their quality of life. Research indicates a positive correlation between higher education levels and an improved quality of life in the elderly [22,23]. Education equips individuals with additional knowledge and information, fostering the adoption of healthier lifestyles and contributing to overall well-being and positive aging. Notably, engaging in educational programs for seniors has been linked to a higher quality of life, particularly among older women [24].

In a previous study, in elderly adults aged 70–90 years, obesity was linked to a decline in overall quality of life, particularly impacting independent living, social relationships, and the perception of pain [25]. Similar results were found in another study in Jiangsu, China, that underweight is an explicit risk factor for low HRQoL in both elderly males and females, while the effect of being overweight on low HRQoL varies slightly by gender [26]. In this study, the findings closely aligned with those of other studies. BMI significantly correlated with QoL, with a lower BMI indicating a lower QoL.

A study in Australia showed that short and long sleep were significantly associated with poor self-rated health and lower quality of life in a large sample of middle-aged and older Australian adults [27]. Another study in China showed that appropriate sleep duration and good sleep quality benefit quality of life. Meanwhile, maintaining good sleep status also reduces the prevalence of depression in the elderly [28]. Our study was consistent with previous studies showing that better sleep duration significantly and positively impacted higher QoL scores [27].

### Limitations

As this was a cross-sectional study, we could only demonstrate a statistically significant relationship between QoL and other predictor variables. However, a causal association cannot be established from this type of study. A cross-sectional study may also suffer from reporting bias (underreporting or overreporting). Furthermore, this study may not be representative of the entire population, as it relies on a single sample at one point in time.

## 5. Conclusions

In Bangladesh, aging has become a significant social issue in the new millennium. To guarantee an overall healthy life for the elderly within the community, the results of this study should benefit policymakers and medical professionals by providing extra attention and enforcing policies for the health of the elderly, the majority of whom were found to have very poor QoL. Targeted interventions are urgently needed to address the poorer QoL of females in general by formulating policies to ensure adequate treatment facilities, early institutional care, and other opportunities for society’s most vulnerable population. Both the government and non-governmental organizations need to offer comprehensive services beyond medication. This may include nutrition care, social support, psychological support, and rehabilitation. Future research should focus on specific service-based interventions that enhance the quality of life in community settings as well as in elderly care facilities.

## Figures and Tables

**Table 1 diseases-12-00212-t001:** Sociodemographic characteristics of the participants (*n* = 275).

Characteristics	All Participants (*n* = 275)	Male (*n* = 173)	Female (*n* = 102)
Age (mean ± *SD*)	65.07 ± 5.5	65.78 ± 5.87	63.85 ± 4.49
Marital status (%)			
Married	215 (78.2)	156 (90.2)	59 (57.8)
Unmarried	2 (0.7)	1 (0.6)	1 (1.0)
Widow/divorced	58 (20.3)	16 (9.2)	42 (41.2)
Education (%)			
No formal education	115 (41.8)	54 (31.2)	61 (59.8)
Primary education (grades 1–5)	53 (19.3)	35 (20.2)	18 (17.6)
Secondary education (grades 6–12)	52 (18.9)	35 (20.2)	17 (16.7)
Higher Secondary	15 (5.5)	13 (7.5)	2 (2.0)
Graduation	26 (9.5)	23 (13.3)	3 (2.9)
Post-graduation	14 (5.1)	13 (7.5)	1 (1.0)
Occupation (%)			
Retired	39 (14.2)	36 (20.8)	3 (2.9)
Farmer	22 (8.0)	22 (12.7)	0
Businessman	23 (8.4)	21 (12.1)	2 (2.0)
Service holder	36 (13.1)	34 (19.7)	2 (2.0)
Unemployed	55 (20.0)	51 (29.5)	4 (3.9)
Housewife	88 (32.0)	0	88 (86.3)
Others	12 (4.4)	9 (5.2)	3 (2.9)
Family monthly income (Taka) ^a^			
Below 20,000	72 (26.2)	42 (24.3)	30 (28.4)
20,000–39,999	135 (49.1)	90 (52.0)	45 (44.1)
40,000–59,999	53 (19.3)	31 (17.9)	22 (21.6)
60,000 and above	15 (5.5)	10 (5.8)	5 (4.9)
Monthly allowances received (%)			
None	206 (74.9)	124 (71.7)	82 (80.4)
Pension	33 (12.0)	28 (16.2)	5 (4.9)
Freedom fighter allowance	9 (3.3)	5 (2.9)	4 (3.9)
Old age allowance	25 (9.1)	14 (8.1)	11 (10.8)
Disability allowance	2 (7.0)	2 (1.2)	0
Residence type (%)			
Urban	131 (47.6)	91 (52.8)	40 (39.2)
Sub-urban	32 (11.6)	21 (12.1)	11 (10.8)
Rural	112 (40.7)	61 (35.3)	51 (50.0)

^a^ 1 US dollar = 109.39 Taka (in June 2024).

**Table 2 diseases-12-00212-t002:** Characteristics of physical health, dietary pattern, sleep hours, and the quality of life of the participants (*n* = 275).

Characteristics	All Participants (*n* = 275)	Male (*n* = 173)	Female (*n* = 102)
Body mass index, mean ± *SD*	24.36 ± 3.32	24.24 ± 2.97	24.56 ± 3.85
Underweight	4 (1.5)	2 (1.2)	2 (2.0)
Normal	170 (61.8)	108 (62.4)	62 (60.8)
Overweight	83 (30.2)	56 (32.4)	27 (26.5)
Obese	18 (6.5)	7 (4.0)	11 (10.8)
Eat vegetables			
Daily	109 (39.6)	70 (40.5)	39 (38.2)
Often	99 (36.0)	61 (35.3)	38 (37.3)
Sometimes	64 (23.3)	39 (22.5)	25 (24.5)
None	3 (1.1)	3 (1.7)	0
Eat fruits			
Daily	33 (12.0)	22 (12.7)	11 (10.8)
Often	42 (15.3)	24 (13.9)	18 (17.6)
Sometimes	171 (62.2)	109 (63.0)	62 (60.8)
None	29 (10.5)	18 (10.4)	11 (10.8)
Eat fish, meat, eggs, or lentils			
Daily	109 (39.6)	67 (38.7)	42 (41.2)
Often	83 (30.2)	51 (29.5)	32 (31.4)
Sometimes	81 (29.5)	55 (31.8)	26 (25.5)
None	2 (0.7)	0	2 (2.0)
Drink milk or milk products			
Daily	57 (20.7)	39 (22.5)	18 (17.6)
Often	36 (13.1)	21 (12.1)	15 (14.7)
Sometimes	123 (44.7)	78 (45.1)	45 (44.1)
None	59 (21.5)	35 (20.2)	24 (23.5)
Smoking status			
Current smoker	30 (10.9)	30 (17.3)	0
Former Smoker	56 (20.4)	56 (32.4)	0
Never smoked	189 (68.7)	87 (50.3)	102 (100.0)
Sleep duration per day			
3–5 h	89 (32.4)	56 (32.4)	33 (32.4)
6–7 h	142 (51.6)	90 (52.0)	52 (51.0)
8–9 h	44 (16.0)	27 (15.6)	17 (16.7)
Quality of Life (QoL), mean ± *SD*	113.79 ± 16.50	116.41 ± 16.23	109.33 ± 16.06
Quality of Life (QoL) score categories			
Bad as can be (less than 99)	60 (21.8)	28 (16.2)	32 (31.4)
Bad (100–119)	119 (43.3)	75 (43.4)	44 (43.1)
Optimum (120–139)	76 (27.6)	54 (31.2)	22 (21.6)
Good (140–159)	20 (7.3)	16 (9.2)	4 (3.9)

**Table 3 diseases-12-00212-t003:** Correlation between quality of life (QoL) scores with sociodemographic factors and sleep duration (*n* = 275).

Sociodemographic Variables and Sleep		QoL Scores
Age	*r*	−0.128
	*p*	0.034
Gender (0 = male, 1 = female)	*r*	−0.208
	*p*	<0.001
Marital status (0 = married, 1 = other)	*r*	−0.178
	*p*	0.003
Education (0 = no formal education, 1 = any education)	*r*	0.337
	*p*	<0.001
Smoking status (0 = never smoker, 1= smoker)	*r*	−0.065
	*p*	0.282
Sleep duration (h)	*r*	0.195
	*p*	0.001

**Table 4 diseases-12-00212-t004:** Factors predicting quality of life in elderly people.

Variables	*β*-Coefficient	*SE*	*p*-Value	95% *CI* for *β*
Constant	128.391	12.145		
Age	−0.408	0.168	0.016	−0.737 to −0.078
Gender (0 = male, 1 = female)	−9.202	2.301	<0.001	−13.733 to −4.671
Educational status (0 = no formal education, 1 = any education)	8.298	1.927	<0.001	4.504 to 12.093
Smoking status (0 = never smoker, 1 = smoker)	−7.426	2.272	0.001	−11.90 to −2.953
Sleep duration (h)	2.101	0.696	0.003	0.731 to 3.471
Marital status (0 = married, 1 = others)	−0.078	−1.307	0.192	−0.08 to 0.833

## Data Availability

The data collection questionnaires are provided in Appendix A and Appendix B. The data presented in this study are available on request from the corresponding author.

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
