# Peer review of "Health-Related Quality of Life among Elderly Patients in Urban Bangladesh: A Cross-Sectional Study"

_diseases, 2024, doi:10.3390/diseases12090212_

Round 1
Reviewer 1 Report
Comments and Suggestions for Authors
This is a well written and well performed study. The presentation is clear, but overwhelmed by statistics of little interest and relevance.
In Table 1, the number of participants with various characteristics is presented for all participants and for males and females, with the percentage distribution. This is fine, and sufficient for the presentation of the study cohort.
In addition, chi-square and p values are presented. These are answers from statistical analysis, but what are the questions? What they tell, is that the probability that the unequal distribution of marital status in men and women in the study group arose by chance from an equal distribution among elderly pasitent visiting the outpatient department is less than 1:1000. And corrrespondingly for education and the other variables. As far as I can see, these analyses are of lttle interest, they are hardly mentioned in the text, are not used in further analyses or mentioned in the discussion and conclusion .
So my suggestion is that the chi-square and p-value columns, as well as the footnotes, are deleted from Table 1. For clarity, I also suggest that in the first column, the characteristics are better distinguished from the specifications by indentation and/or different letter type:
Age (mean...
Marital..
Married
Unmarried
Widow/..
Education (%)
No formal..
I suggest same deletions of chi-square and p columns and typographical changes in Table 2
Figure 1 adds nothing to the values listed in Table 2 and can be taken out.
In Table 3, the only data of interest are those of the last column, the bivariate relationship between the sociademographic factors and Qol scores, which are the basis for the following multivariate analysis. The rest of Table 3, the bivariate relationship between all sociodemographic factors, is not mentioned in the text, are not used in further analyses, and has no place in discussion or conclusion. These 7 columns can be deleted.
In Table 4, the non-significant values for age and gender could be included in the table, rather than listed as excluded in a footnote (?)
As a consequence of the deletions suggested above, the following changes are suggested in the text:
Lines 23-24: Delete 'The Chi-square..........variables'
128-130: Delete: 'To examine............was used'
139-140: Delete: '(95% confidence...........3,17)'
141-143: Reformulate to: ' The majority of the participants were male (62.9%), married (78.2%), and had no formal education (41.8%) or primary education only (19.3%) (Table 1).
The appendixes are best taken out of the main report and presented as Supplemental material.
With these changes, the report can be accepted as a well written and more readable presentation.
Reviewer 2 Report
Comments and Suggestions for Authors
Aging is a natural biological process and an existential human experience. The challenge is how to use free time wisely and prudently after completing professional duties, and consequently gain joy, knowledge and satisfaction from overcoming one's own weaknesses and adversities. For a person who considers themselves strong, important and significant, the world is something safe, familiar, allowing one to manage oneself even when experiencing certain illnesses and ailments. For someone who feels useless and weak, the environment changes into an arena for the action of hostile forces, a source of threat and fear, and the sense of the quality of one's life deteriorates. These ideas and attitudes are strongly culturally and even historically conditioned. The quality of life in old age is determined by many interrelated factors, among which health is given an important place. For this reason, the authors' undertaking of a study on the quality of life among the elderly was fully justified.
Reviewer 3 Report
Comments and Suggestions for Authors
This is a well-structured article with a solid study design and thorough analyses. Here are few suggestions that might help strengthen the conclusions.
1. The association between age, gender and QoL was one of the key findings of this article. It would be nice to also include a regression model with age and gender included, or alternatively, present stratified models by gender. If the stratified analysis is planned, it might also be interesting to also include smoking status in the full model.
2. In line 205, the authors mentioned that "there was no multicollinearity observed among the independent variables". However, Table 3 showed correlations between variables such as age and gender, education and marital status, which seems contradictory. Please revisit this point.
3. The inclusion of educational status and marital status as continuous variables in the regression model raises some questions. While it makes some sense for the educational status (still, this approach assumes equal difference between levels, e.g., no vs. primary, higher secondary vs. graduation, which may not be accurate), it's odd to do so for marital status.
It would be more appropriate to use dummy variables for different marital status (and potentially education as well) in the multivariable regression. Given that only 2 participants were unmarried, it's also fine to use a binary variable, married vs. others.
4. I think there was a typo in table 1 for married females, the reported percentages seem higher than expected. Please double-check the result tables for accuracy.
